# EmoAI Smart Classroom: The Development of a Student Emotional and Behavioral Engagement Recognition System

## Abstract

Emotions significantly influence the learning environment, impacting student engagement and the overall educational process. With the rise of challenges in maintaining student engagement in large offline classrooms due to various factors, there's an increasing need to detect classroom emotions. This study aims to detect and analyze emotions evoked in classrooms to enhance the educational experience for both educators and learners. Utilizing the DAiSEE dataset, which captures various affective states in offline classroom settings, an engagement detection system was developed using the YOLOV8 state-of-the-art model and deployed on Roboflow. The system's framework was based on capturing facial cues and was augmented with an interactive interface for lecturers. Despite its advanced capabilities, the model achieved a precision of 74.5%, a recall of 60.6%, and a mean average precision (mAP) of 65.3%. The findings suggest that while the model offers significant insights, there's potential for further refinement, particularly given the limited frames used for training. The study's interactive interface offers real-time feedback for lecturers, underscoring the intertwined relationship between emotions and learning. Future directions include real-time engagement detection and alert systems, emphasizing the potential to revolutionize classroom dynamics through emotionally attuned educational environments.

**keywords:** emotion, emotion recognition, engagement, classroom, affective computing, camera, Roboflow, YOLO

## 1 Introduction

According to Hampton (2006), Emotion is a specific sensation or feeling in the mind that provides directional drive to the other faculties of the mind - memory, intelligence, and physical activities - for their actions to be performed to pursue a specific goal. Emotions and learning are closely intertwined, and they play a significant role in creating the learning environment.Krithika & GG (2016) Emotions can have an impact on how pupils learn Meyer & Turner (2002), it may have an impact on a student's concentration, drive to learn, coping mechanisms, and self-control Furrer et al. (2014). It may also provide crucial cues regarding how well they are studying in class. Understanding a student's emotional state is crucial for a teacher, even though it might be challenging to measure emotions Alkabbany et al. (2019).

Different emotions are evoked in a classroom setting, which might influence whether learning occurs or not. There is some evidence that some emotions, such as boredom Csikszentmihalyi (1990); perplexity (D'Mello et al. (2006); Aist et al. (2002)), and flow (also known as engagement, Csikszentmihalyi (1990), and frustration, might affect learning and cognition (Aist et al. (2002).

The issue of student disengagement is currently getting worse every day due to a variety of factors, including poor teaching methods, a limited attention span, and a lack of student-teacher interactions Bradbury (2016) Lamba et al. (2014). The presence of big offline classrooms (student count is greater or equal to 60) exacerbates this issue. In order to improve the educational system, student engagement—which occurs when a student participates meaningfully in the learning environment—should be carefully considered Sharma et al. (2019)

In this paper, we try to detect the emotions that are evoked in the classroom. The engagement detection system was built using the YOLOV8 state-of-the-art model and was deployed on Roboflow. The model achieved 74.5% precision, 60.6% recall, and 65.3% mAP.

## 2 REVIEW OF RELATED WORKS

In a study by Pabba & Kumar (2022), a vision-based automated system for student group engagement in a large offline classroom environment was developed by analyzing students' academic affective states through facial expressions. The facial expression dataset used was created from classroom lecture video frames of more than forty students. A Convolutional Neural Network-based architecture was used as the framework for the recognition system. Multi-modal analytics went a step higher in a research conducted by Peng & Nagao (2021) where they monitored students' mental states (boredom, frustration, concentration, and confusion) during classroom discussions. An intelligent multi-sensor system was developed using modalities such as facial, heart rate, and acoustic indicators. A set of machine learning algorithms such as support vector machine, random forest, and multilayer perceptron were trained using features from facial, heart rate, and auditory modalities. However, the implementation of this study is expensive because it needs physical devices such as Apple Watch, and Air pods to measure student's multi-modality data which is costly for large offline classroom settings. Zheng et al. (2020)identified students' engagement by analyzing behaviors detected from the students. An improved Faster R-CNN model was trained on the created classroom students' behavioral dataset consisting of hand raising, standing, and sleeping behaviors. However, this study is constrained to only students' behaviors with no room for academic-based emotions for estimating students' engagement. The research conducted by Sharma et al. (2019)introduces a multi-camera-based emotion detection system in a classroom environment. The system is able to detect and record changes in students' facial expressions and report to the teacher in real-time. Three methodologies (Viola Jones, Gabor Wavelet, and Multi-Layer Perceptron) were combined to dynamically recognize the students' basic emotions in real-time.

## 3 METHODOLOGY

### 3.1 Data Collection

DAiSEE (Dataset for Affective States in E-Environment) is the first multi-label video classification dataset comprising 9,068 video snippets captured from 112 users for recognizing user affective states of boredom, confusion, engagement, and frustration in the wild. the dataset has four levels of labels namely: very low, low, high, and very high for each of the affective states The DAiSEE dataset was collected from a link provided in an article published on Paper with Code.

### 3.1.1 Why DAiSEE?

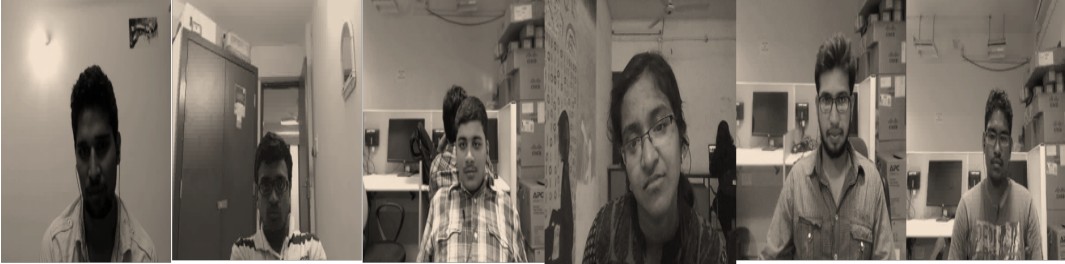

Fig i. Screenshot of DAiSEE

The choice of DAiSEE for building the engagement system is a result of the following reasons:

(i) It is the first publicly available dataset for studying user engagement and related affective states.

(ii) DAiSEE presents a dataset to understand more subtle states that are not explicitly exhibited in humans.

(iii) It contains videos that allow researchers to use temporal information for effective recognition

### 3.2 Frame Extraction

In order to capture the temporal changes in affect, a tool known as FFMPEG was used to extract frames at 30 frames per second (30fps). The frames extracted from the video (9,068) totaled 2,723,882.

### 3.3 Data Annotation

707 images from the frames were loaded into the Annotation section of RoboFlow and bounding boxes were drawn around each extracted frame with its corresponding emotion class. The classes were: Confusion, Boredom, Engaged, and Frustration. This process was done using Label Assist on Roboflow (see fig ii). The labeled data then formed the data for model building

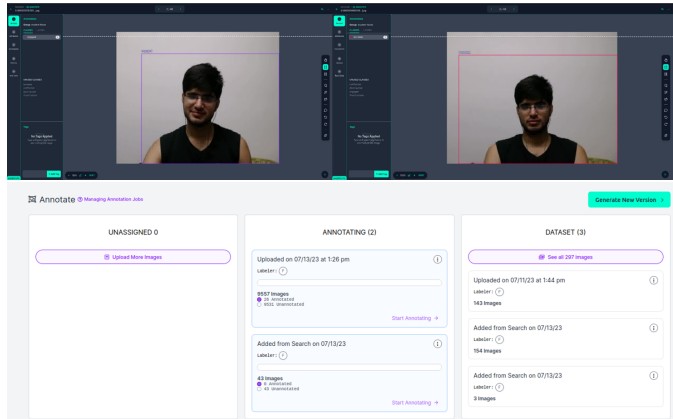

Fig ii. Screenshot of Data Annotation process on Roboflow

### 3.4 Model Building

This talks about the data pipeline from splitting to making inferences on Roboflow. The stages involved are explained below:

#### 3.4.1 Data Split

To prepare the dataset for benchmarking, we create a data split into train, validation, and test sets. For medium datasets, the 80-10-10 ratio is used due to its effectiveness in balancing the needs of training, validation, and testing Hastie et al. (2009). (see fig iii)

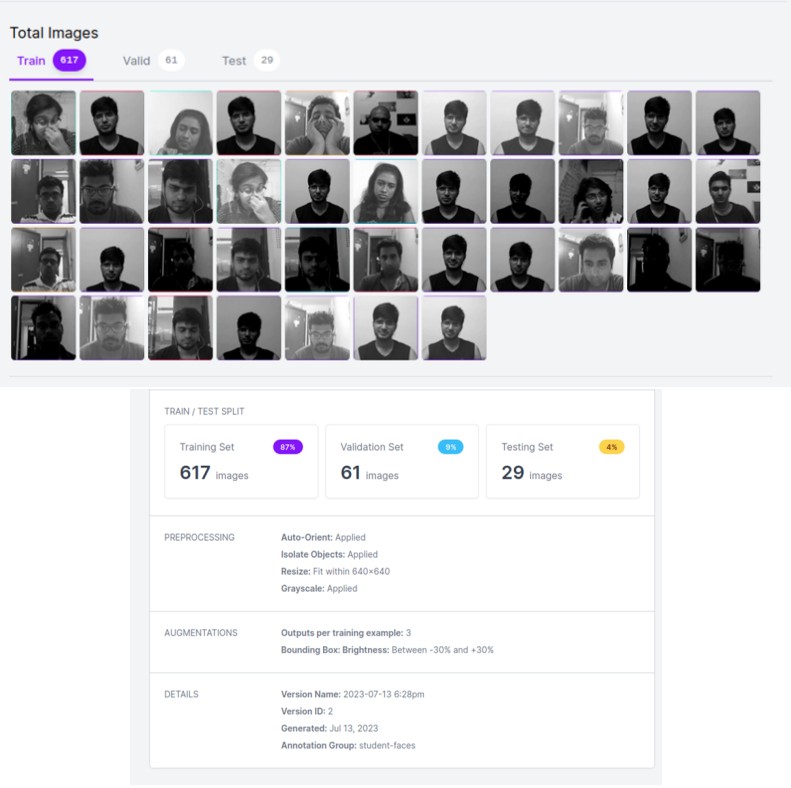

Fig iii. Screenshot of Data Splitting process on Roboflow

### 3.4.2   Data Preprocessing

To further prepare the dataset for modeling, the following were used to enhance the quality and suitability of the input data for the detection model.

- Grayscale Conversion: The images were converted to grayscale to reduce the color information and to make it easier for the model to focus on object shapes and edges.

- Resizing: The images were resized to a consistent resolution (624X624) to maintain uniformity in the input data. Doing this makes it easier for the model to learn and generalize across different objects.

- Auto-Orient Correction: The images were made to have the same orientation (portrait). This helps the model learn object features consistently without being confused by variations in rotation.

- Bounding Box Brightness Adjustment: The brightness of the regions within the bounding boxes was adjusted to 70% to make it easier for the model to detect images accurately.

### 3.4.3   Model Training

The object detection model was trained by building upon the foundation of a pre-trained YOLO V8 architecture. The pre-trained model's weight was fine-tuned in the pre-processed dataset. During training, the dataset was fed into the model which learns to refine its internal features and parameters. The training time took one hour (1hr) to be completed and for class loss results to be generated.

### 3.4.4   Evaluation Metrics

To assess the performance of the object detection model, the following metrics were used and the results are indicated.

- Mean Average Precision (mAP): Mean Average Precision is a measure that evaluates the accuracy of object detection by considering both precision and recall. It computes the average precision across different levels of confidence thresholds for object detection. The model recorded 65.3% mAP which indicates that on average, the model's predicted object bounding boxes are correct for about 65.3% of the objects in the dataset.

- Precision: Precision is the ratio of true positive predictions (correctly detected objects) to the total number of positive predictions made by the model. The model recorded 74.5% precision which indicates that out of all the objects the model predicted as positive, approximately 74.5% were actually correct.

- Recall: It is also known as sensitivity or true positive rate. It measures the model's ability to correctly detect a certain percentage of all the actual objects in the dataset. The model recorded 60.6% recall which means the model was able to detect around 60.6% of all the actual objects present in the dataset.

**Benchmark Result** The multi-level classification approach presented in Solanki & Mandal (2022) achieved a mean accuracy of 0.7741 across all metrics which surpassed the single-frame image model presented in this paper which achieved accuracies ranging from 45% to 65% for various video evaluations.

The result of the model training gave 74.5% precision with the following loss results (see fig iv).

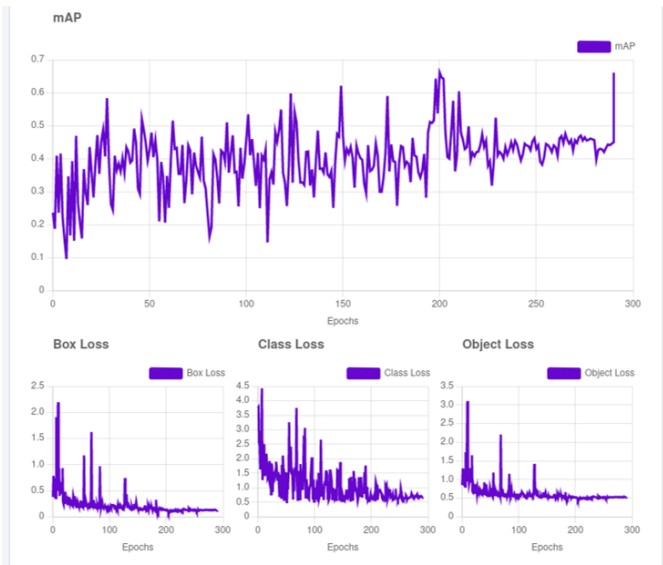

Fig iv. Screenshot of Class loss results of the model on Roboflow

### 3.4.5 Inference

The system was tested by uploading pre-recorded classroom videos. The result is shown in the figure below:

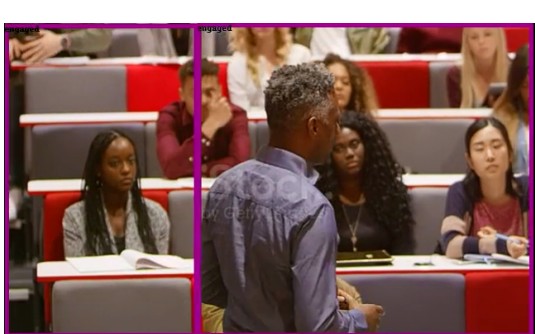
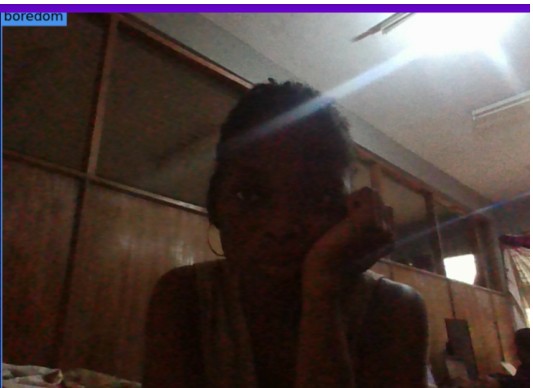
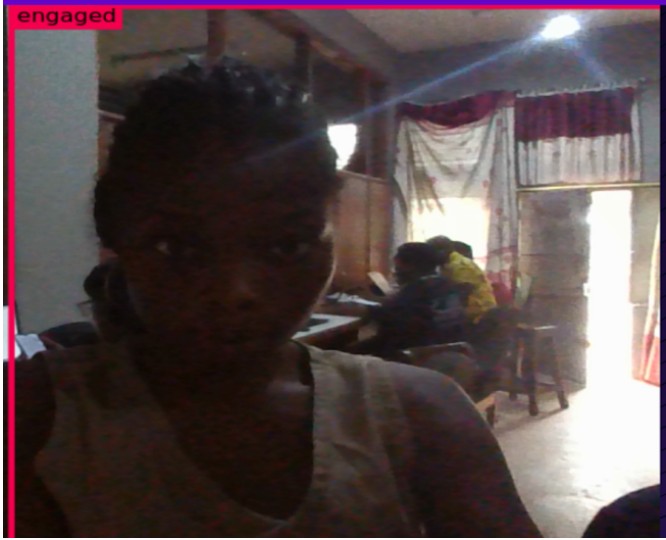

Fig v. Screenshot of model inference on classroom video

## 4   Model Deployment

The following are the steps taken to connect the model with the interface. The interface that was built was the lecturer's part where the lecturer as an admin uploads the video to get an engagement analysis report.

- Step 1: Developing the ML Model and API

  1. The model was made to classify the engagement into three levels: High, medium, and low.
  2. Next, an API endpoint is established within the backend infrastructure. This endpoint, created using a programming language like Python and frameworks such as Flask or FastAPI, exposes the ML model's classification capabilities.
  3. The endpoint's functionality encompasses accepting uploaded classroom videos, performing any necessary preprocessing, and then passing the videos through the ML model. Subsequently, the endpoint returns the determined engagement classification, thus forming a bridge between the backend and the model.

- Step 2: Setting Up Strapi CMS

1. To manage content, a Strapi CMS is installed and configured on the server (see fig vi). This involves defining requisite content types and corresponding fields. One key content type, labeled "Video," includes fields like title and a mechanism for uploading video files.

2. Role-based permissions are tailored to allow lecturers to upload videos, while administrators can oversee content management.

3. In the Strapi CMS, a webhook or server function is implemented to activate whenever a new video is uploaded. This function serves to connect with the ML model's API endpoint, passing the uploaded video for engagement classification. The obtained classification report and video details are subsequently stored in the Strapi database.

- Step 3: Developing the Next.js Front End

  1. The process begins by configuring a Next.js project for the frontend.

  2. A user-friendly interface is crafted, enabling lecturers to seamlessly upload classroom videos. Within this interface, user-friendly form components are integrated, facilitating video detail input and file uploads.

  3. Upon uploading a video, the Next.js application initiates an API call to the Strapi server. A loading indicator visually represents the ongoing process.

  4. Once the engagement classification report is generated, the interface dynamically presents both the uploaded video and the associated engagement classification, providing a comprehensive overview of the uploaded content

- Step 4: Integration Flow and User Interaction

  1. Lecturers access the application and securely log in.

  2. Within the Next.js application, a designated section facilitates the hassle-free upload of classroom videos.

  3. Lecturers select the desired video file.

  4. By triggering the "Upload" command, the Next.js application seamlessly transmits the video file and its details to the Strapi backend through an API call.

  5. Strapi, upon receiving the data, promptly activates a webhook or server function, which establishes communication with the ML model's API endpoint. The uploaded video is then sent for engagement classification.

  6. After processing, the ML model's classification report is communicated back to Strapi, which dutifully stores both the video details and the engagement report within its database.

  7. A subsequent API call from the Next.js application to Strapi retrieves the engagement report, categorizing engagement as low, medium, or high.

  8. The resulting engagement report (see fig vii)is visually presented on the user interface, providing lecturers with valuable insights into the uploaded classroom video's engagement levels.

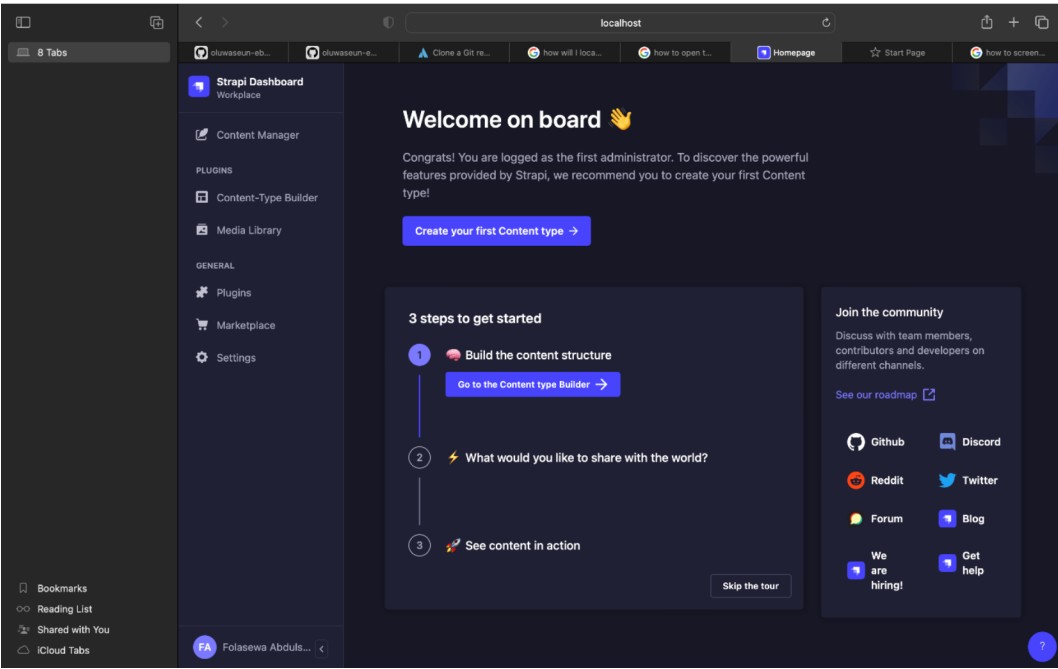

fig vi. Interface of the Strapi Dashboard

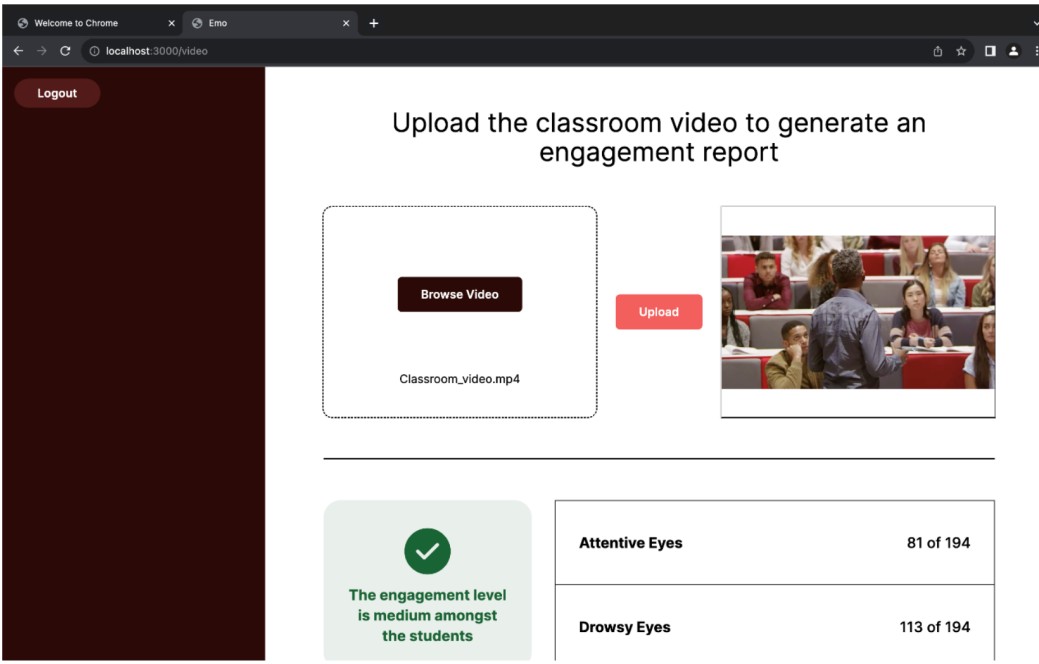

fig vii. The Engagement report dashboard

## 5    System Evaluation

The system was evaluated using alpha testing. The choice of this form of evaluation is the fact that the system has not been deployed for end users to use. It is just available on the localhost computer for the developer to test and evaluate.

### 5.1   Parameters for Evaluation

The following are the parameters used to evaluate the performance of the web app:

- Accuracy and Precision: This is used to measure how accurately the system detected engagement levels in the video.

- User Interface and Experience: This is used to measure how user-friendly the interface is for uploading videos and viewing results.

- Response Time: This is used to measure how quickly the system processes and analyses the uploaded videos.

- Compatibility: This is used to measure how well the system works across different browsers.

- Reliability Across Settings: This is used to test the system's performance in various classroom settings, lighting conditions, and video quality.

  The system was evaluated using a measurement scale of Excellent, Good, and Bad. The table1 shows the result of the evaluation after testing of each video.

Table 1: Result of Evaluation

| Video | Length | Accuracy | Ease | Response | Reliability | Compat. |
|-------|--------|----------|------|----------|-------------|---------|
| video_1 | 00:25 | 56% | Good | 21s | Good | Chrome |
| video_2 | 00:14 | 65% | Good | 10s | Good | Chrome/Edge |
| video_3 | 00:46 | 45% | Good | 32s | Bad | Chrome/Edge |
| video_4 | 00:16 | 60% | Good | 15s | Good | Chrome/Edge |

**How does the system perform using these metrics?**

- Accuracy vs. Reliability: There appears to be a correlation between accuracy and reliability. For instance, Video3, which has the lowest accuracy at 45%, also has a "Bad" reliability rating. The other videos, with higher accuracy values, have "Good" reliability ratings.

- Response Time vs. Reliability: Video3, which is rated "Bad" for reliability, has the longest response time of 32 seconds. This suggests there might be a correlation between longer response times and reduced system reliability.

- Ease of Use: This metric remains consistent as "Good" across all videos, so it does not offer a discernible correlation with the other metrics in the table.

- Compatibility: This metric is more about the system's ability to work across different browsers and doesn't correlate directly with other performance metrics like accuracy or reliability.

  In summary, there seems to be a potential correlation between accuracy, reliability, and response time. When the system's accuracy decreases, its reliability might also suffer, and the response time could increase. However, more data points or deeper statistical analysis would be required to confirm these correlations definitively.

## 6   Summary and Conclusion

In this thesis, we present DAiSEE because of the novelty it comes with by having different affective states such as boredom, frustration, confusion, and engagement. The choice of the dataset for this thesis is because it captures the nature of students learning in an environment with varying poses and positions. Although the state-of-the-art model was used to train the dataset, we expected a higher precision due to that, but, the precision achieved was far less than the previous work upon which this thesis was based. The observation we could make is due to the few extracted frames used to build the model. We also made an attempt to

give the model an interface where lecturers can interact with to get the engagement analysis of the students. This could be considered an edge over the existing engagement system and if proper research is further done, the system can be scaled up for real-time implementation. In conclusion, emotions impact learning and subsequently impact engagement and attention and if research is properly done in solving the challenges and limitations, student's affective states will be better understood, lecturers will be able to dynamically change the direction of lectures and learning would not be emotionally exhaustive for both the learner and the tutor.

## 7 CONCLUSION AND FUTURE WORK

Emotions are much more complex Keltner et al. (2019), and their multidimensional nature remains a great challenge for future work. Most especially estimating student engagement. Lots of work has been done on estimating student engagement in large classrooms through automatic methods that use computer vision techniques. While some research preferred to use only facial expression cues in estimating student engagement, some research preferred to use behavioral cues. The belief is facial cues tend to be less occluded in relation to body cues like body posture. In this work, we presented the DAiSEE dataset to develop an engagement detection system. The unique proposition is the fact that it is intended for offline classrooms. We hope that this system will be the third eye for tutors in the classroom while lectures are being delivered without being invasive. This work, based its working architecture on the work done by Pabba & Kumar (2022). However in place of the Convolutional Neural Network used, YOLO V8, which is the current state of the art was used. Although over two million frames were collected, while building the system, seven hundred data were used as an initial proof of concept. The environment provided for evaluation is one in which the tutor uploads a pre-recorded classroom video after the lecture. One of the limitations of the system is that the system can not recognize faces wearing glasses. Going forward, the system can be upgraded to work in real-time to give alerts when it detects the engagement level of the students is low or below the threshold. Tutors can be more aware of the atmosphere in the classroom which in turn can help improve the dynamics of lectures delivered. Improving the accuracy of the model while decreasing computational load is worth giving attention to. This could be seen in improving the robustness of the dataset to accommodate students from different ages, cultures, skin color,s and gender. The system could be further improved to enable real-time detection of engagement with the inclusion of an alert system when the students fall below an engagement threshold.

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
