# OpenReview forum: "EmoAI Smart Classroom: The Development of a Student Emotional and Behavioral Engagement Recognition System"
_TMLR — Rejected by TMLR_

### Review · Reviewer_G1Cf · 2024-01-18

**Summary Of Contributions:**

This paper presents a system to monitor emotions and engagement in classroom settings using a fine-tuned YoloV8 classification model. This work may contribute to work developing AI systems for lecturer feedback in educational environments and may reveal insights on the relationship between emotions, engagement, and learning within the classroom.

**Audience:**

Yes

**Broader Impact Concerns:**

No specific concerns.

**Claims And Evidence:**

No

**Requested Changes:**

1. (3.2, 3.3) For data collection, the paper uses 707 frames from DAiSEE’s 2 million frame dataset. Please clarify how these frames were chosen. Randomly? From one video or multiple videos?
2. (3.3) Clarify the data annotation more.
* The extracted frames from the dataset were annotated in RoboFlow with four emotion classes. Aren’t these frames already annotated by the DAiSEE dataset with these emotions. Why were they re-annotated?
* Clarify the difference between your annotation and the original annotation. The original dataset was a multi-label classification task with each emotion ranging from low to high. Was high to low used in the annotation of four emotion classification on RoboFlow?
* Mention the number of annotators, whether annotator agreement was used, what was the qualifying criteria to annotate the different emotions
3. (3.4.4) Training evaluation metrics should be clearer.
* I suggest putting the metric results in a table, so that they are easier to read
* Clarify whether the performance metrics that you report are for the training set, validation set, or test set.
* It would be nice to have a breakdown of performance on the different emotion classes
* If possible, include more baselines for comparison.
4. (3.4.5) Inference: it is unclear where the pre-recorded classroom videos are from. Please provide more description of this data: how was it collected, how much data was collected, whether the data was annotated or if this inference is qualitative.
5. (4) Model deployment step 1 claims that the model was made to classify engagement into three levels; however, from my understanding, the annotation on RoboFlow was a classification task for four different emotions without levels. How did you determine the levels of engagement?
6. (5) Please clarify details on system evaluation
* Specify the details of the videos you evaluated on: how they were collected and how much was collected, were the videos annotated?
* Adjust the columns on the Table 1 to exactly reflect the metrics you describe (e.g., “Ease” equates to “User Interface and Experience”? Is “Accuracy” reporting accuracy or precision?)
* Describe the annotator evaluation in more detail: how many annotators, did they have agreement, what was the qualifying criteria for the different scales of Excellent, Good, and Bad.
7. Minor changes:
* Fix citations and names of figures to match specified TMLR format (e.g., citations without parentheses around them, figures use roman numerals rather the digits, etc.)
* Some citations missing, such as citation for the dataset you use (DAiSEE)

**Strengths And Weaknesses:**

Strengths:
* Presents a simple, easy-to-use AI system to monitor emotions in the classroom.
* Because the system only uses video modality, it is less invasive and less expensive than prior work monitoring engagement that may use more complex signals requiring SOTA wearable sensors (e.g., Apple Watch, heart rate monitors, microphones, etc.)

Weaknesses:
* The authors claim that emotions are a reliable indicator for student engagement within the classroom. However, it is unclear if solely emotion detection is a reliable signal for engagement. For example, even if a student has confused emotion, they may still be engaged and learning. Additionally, resting facial expression detected by a computer vision model may not reflect how the student is actually feeling. While the authors intend to supply evidence that emotion can reflect engagement in related works, much of this work uses multiple signals, such as heart rate, auditory modalities, and behavioral actions. More evidence may be needed to show that solely emotion detection is a reliable indicator for engagement or learning in the classroom.
* The paper does not provide alternative baselines to compare the system (besides Solanki & Mandal system which uses a binary classification task, which seems different from the paper's multi-class classification approach?).
* There is lack of clarity in the details and metrics within the dataset & annotation, training, and evaluation, which makes it difficult to understand the effectiveness of the entire system.
* See requested changes for more details.

---

### Review · Reviewer_Yz2p · 2024-01-19

**Summary Of Contributions:**

The authors create a system to detect emotions and engagement in offline classrooms, which could be used to assist lecturers. They extract and annotate a few frames from the DAiSEE dataset, and then fine-tune an object detection model. This model is then integrated into a web application for the lecturer to obtain an engagement report about its students.

**Audience:**

No

**Broader Impact Concerns:**

The authors would ideally discuss potential negative impacts of their technology, and how to mitigate such effects. For example, we wouldn't want students to be penalized by their teachers if they look disinterested according to the model.

**Claims And Evidence:**

No

**Requested Changes:**

**Critical**

- Cite DAiSEE properly (A Gupta, A D'Cunha, K Awasthi, V Balasubramanian, DAiSEE: Towards User Engagement Recognition in the Wild, arXiv preprint: arXiv:1609.01885). You can also cite the GitHub repo for YOLOv8.

- Clarify or correct this statement in the abstract: "the DAiSEE dataset, which captures various affective states in offline classroom settings". The "EE" in the acronym mean E-Environments, which is not an offline classroom setting.

- Revise sections 6 and 7. There shouldn't be 2 conclusions.

**Important**

- Cite more recent work to claim "The issue of student disengagement is currently getting worse every day".

- In figure v, you might not want to use a stock photo with the watermark still on.

**Typos and other minor issues (might not be exhaustive)**

You could briefly describe Roboflow as readers may not know what it is.

Page 1: Emotion→emotion

Move period after citation of Krithika & GG (2016).

Page 2: "the dataset has four levels of labels namely: very low,
low, high, and very high for each of the affective states": Capital t at the beginning, period at the end.

Air pods→AirPods

Page 3: affect→effect

Page 4: in the pre-processed dataset → on the pre-processed dataset

Page 9: "The table1"→ "Table 1"

Do not refer to your work as a "thesis"

Rephrase "state-of-the-art model was used to train the dataset".

Page 10: Avoid fragment "Most especially estimating student engagement."

Page 11: Remove comma in "This work, based its working architecture"

**Strengths And Weaknesses:**

**Strengths**

- The task is well motivated. Such a system could possibly help a lecturer adapt his presentation to keep the students more engaged, and ultimately improve how well they learn.
- The full system has been implemented and successfully deployed (locally).

**Weaknesses**

- The scope of the paper might not be appropriate for TMLR. It is more a demo/system description than a typical research paper.
- The training data does not seem ideal for the task. Only a few (707) frames are used, without clear scientific justification. Most frames contain a single user (in the foreground), while offline classrooms would likely contain many people (if we only cared about 1 person, classification would likely be preferred to object detection/labeling). Finally, according to the DAiSEE paper, all subjects are Asian, which might introduce bias and lower generalization.
- Evaluation is not sufficient. The baseline description ("Benchmark Result") is unclear and I am not sure if the results are directly comparable (what are the "various video evaluations"?). The full system evaluation is based on only 4 videos.

---

> ### Author Response · Authors · 2024-01-27
> **To: Reviewer Yz2p**
>
> Hello, thank you for taking the time to review my work.
>  Which section should I discuss the negative impacts of the technology discussed? Conclusion? or a separate section?
>
> Then, citing more recent works to support my claim that student disengagement is getting worse. That should come under introduction, right?
>
> Then, I made corrections to the two concluding sections, made appropriate citations, and rephrased my abstract and introduction.
>
> About the training frames, how do I justify the no of frames used? Because I had a limit on the number I could upload on Roboflow and I could not process huge amounts of data on my system due to system lag.
>
> I appreciate the time to proofread my work. I would be anticipating your response (s). Have a great day!

---

> > ### Comment · Reviewer_Yz2p · 2024-02-06
> >
> > Which section should I discuss the negative impacts of the technology discussed? Conclusion? or a separate section?
> >
> > > You could add a section named "Broader Impact Statement", as in this paper (https://openreview.net/pdf?id=4i1MXH8Sle).
> >
> > Then, citing more recent works to support my claim that student disengagement is getting worse. That should come under introduction, right?
> >
> > > Yes, that can fit within the introduction. It would help motivate the paper.
> >
> > About the training frames, how do I justify the no of frames used? Because I had a limit on the number I could upload on Roboflow and I could not process huge amounts of data on my system due to system lag.
> >
> > > Unfortunately, I don't have a clear answer. You can mention that it is because of resource constraints. However, that doesn't address if other tools besides Roboflow could have been more appropriate.

---

### Review · Reviewer_SMar · 2024-01-21

**Summary Of Contributions:**

This work proposes a system named EmoAI Smart Classroom to detect and analyze emotions of students evoked in classrooms using the YOLOV8 detector on Roboflow. The proposed model achieves a precision of 74.5, a recall of 60.6, and a mAP of 65.3% on the DAiSEE dataset.

**Audience:**

No

**Broader Impact Concerns:**

No specific ethical concern is in this work.

**Claims And Evidence:**

No

**Requested Changes:**

Please refer to the weakness above.

**Strengths And Weaknesses:**

<Strengths>
1. This work addresses an important problem – automatically detecting engagement of students in the classroom.

2. This work contributes a large-scale dataset for the task named DAiSEE (Dataset for Affective States in E-Environment).

<Weaknesses>
1. This submission is not a good fit to TMLR.
- While TMLR focuses on novel high-quality machine learning research, this work is a simple implementation for an education application using well-known vision techniques.

2. This work lacks technical novelty.
- Most of this submission covers how to implement the proposed system well-known vision models like YOLOv8.
- There is no meaningful technical novelty and contribution.

3. Evaluation is highly limited.
- This work simply measures the performance of the implemented system.
- There is no comparison with neither state-of-the-art methods nor other possible options.

---

### Decision · Action_Editor_Qwyg · 2024-02-21

**Recommendation:** Reject

**Comment:**

The reviewers do not feel there is a technical contribution in the current work, nor that there is sufficient contribution to the learning sciences.  Future incarnations with broader baselines, experiments, and algorithmic novelty may lead to a sufficient audience.

**Audience:**

Education researchers who aim to use minimally invasive technology to improve education

**Claims And Evidence:**

The creation of a system that uses existing ML techniques to extract emotion relevant features for assessing student engagement.  The result is a system that uses only visual features.  It's simple architecture and there is a basic evaluation/comparison after doing datacollection.